# Small Signal Stability Analysis of a Microgrid in Grid-Connected Mode

Hammad Alnuman 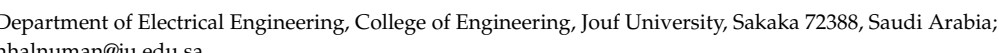

Department of Electrical Engineering, College of Engineering, Jouf University, Sakaka 72388, Saudi Arabia; hhalnuman@ju.edu.sa

**Abstract:** Microgrid stability issues are classified into three categories: transient, voltage, and small signal stability (SSS). Small variations in the load demand and small perturbations in the control system and line impedance parameters can cause instability, which can be avoided by performing an SSS analysis. This paper focuses on investigating the impact of line impedance and passive filter parameters on the stability of a MG in grid-connected mode. Therefore, a MG system was represented mathematically, before performing an SSS analysis that calculated the stability margin of the MG parameters. A sensitivity analysis was performed to determine those parameters highly participating in the SSS. The mathematical results were validated using the simulation results, which were obtained using MATLAB Simulink.

**Keywords:** droop control; microgrid; small signal stability

## 1. Introduction

A microgrid (MG) is a group of micro-sources and loads in a subsystem, which can work in an islanding mode or in a grid-connected mode. If the MG is in grid-connected mode, its frequency and voltage will be dominated by the grid, which makes stability less crucial than that when it is in the islanding mode, where the variation of load demands causes major disturbances to the frequency and voltage, which can lead to destabilization [1,2].

Stability issues in MGs are classified as small signal, transient, and voltage stability. Small signal stability (SSS) is related to the feedback gains of controllers, changes in power demands, and small perturbations in system parameters. A MG is transiently stable if it reaches the steady state condition after large disturbances such as faults, large load steps, and switching to the islanding mode [3].

The power output of renewable energies changes with time, which is considered a perturbation to the system. This power variation makes it challenging for MGs to produce reliable and stable power, especially when there is variation in the load. Therefore, it is crucial to study the impact of small power variations on MG stability, which are considered SSS [4].

SSS can be studied using the eigenvalue theorem, where the system is unstable if at least one of the eigenvalues is on the right half plane of an imaginary axis. However, knowing the eigenvalues is not possible without obtaining mathematical models of the whole system. Therefore, a MG system should be represented mathematically by linear equations, before forming the state space model that allows stability analysis [5,6].

Power demand variations and parameter perturbations are considered SSS issues that can affect the reliability of MGs. Therefore, it is crucial to provide accurate mathematical models for MGs to analyze their stability conditions with respect to small changes to MGs parameters and power demand variations [4]. Power systems behavior is represented by non-linear differential equations. Thereafter, the stability condition can be tested by using eigenvalue analysis, which can be compared with time domain simulations. The

accuracy of the mathematical model representing a MG can be measured by the percentage of matching between the analytical results and time domain results. A system is considered stable if it is able to function again around the same operating point after a disturbance. For example, the voltage of a system is stable if the voltage at all buses is settled after any kind of disturbance. If the voltage becomes unstable, its behavior will be either falling or rising [6,7].

When a MG is in grid-connected mode it is dominated by the grid, which makes it less possible for the MG to be unstable. Stability is a big concern when a MG is in islanding mode, where stability is highly related to control gains, load fluctuations, and line impedance variation [8].

The authors of [9] studied the SSS of a MG in islanding mode, with two distributed generators. The parameters of current, voltage, and LC passive filter were varied in small steps to study the impact on stability. In [10], stability state was examined by applying a load step to a MG in the islanding mode. A mathematical model, simulation model, and experiment setup were provided, which showed the same results. The authors of [11] used SSS analysis to study the impact of droop control gain variation and load variation on the stability of a MG, with three distributed generators in islanding mode. Similarly, in [12] SSS analysis was performed on a MG with two distributed generators in islanding mode, with respect to droop control gains and load variation. The authors of [13] developed a small signal model of a MG in islanding mode, to analyze the stability and robustness of the MG, and then an adaptive control method was proposed to adjust the control gains in response to disturbances. In [14–20] the SSS of MGs in islanding mode was studied when controller gains were increased. In [18], the SSS of a MG in islanding mode was resolved, while proposing a method that simultaneously ensures stability and optimal power flow. Most research related to the SSS of MGs in islanding mode has focused on designing controllers to manage the variation of the load demands, while ensuring stability of the MG system [19].

The authors of [21] used an SSS analysis to examine the impact of droop control gains on the stability of a MG in grid-connected mode. SSS analysis was implemented in [22] to discover the stability margin of control gains in a MG in grid-connected mode. The authors of [23] used an SSS analysis to study the stability of a MG in grid-connected mode, with respect to load demand variations. The authors of [24] derived the SSS limits of a phase locked loop, responsible for synchronizing a wind farm with the grid. The authors of [25] provided a small signal model of a MG with dynamic loads and transmission lines, before investigating the parameter sensitivity and proposing methods to the enhance stability of MGs under different scenarios and configurations.

It is important to study any possible disturbance of MGs, especially in the islanding mode, because the inertia is low and MGs are normally associated with renewable energy sources. It is assumed that MGs are stable when they are in the grid-connected mode, as they will be dominated by the grid. Therefore, previous research has focused on the SSS analysis of MGs in islanding mode with respect to control gains and load demand variations. Only a few articles discussed SSS when the MG is in grid-connected mode, as the MG will be dominated by the grid and small disturbances to control gains and load demands do not normally cause instability. Furthermore, there are scarce published data on investigating the SSS of MGs in grid-connected mode, with respect to LCL filter and line impedance parameter variations. Therefore, it was decided in this article to investigate the SSS of a MG when it is in grid-connected mode.

The main contributions of this article are summarized as follows:

- Deriving a state space model of a MG in grid-connected mode, which involves a LCL filter, line impedance, and control system.
- Investigation of the impact of the LCL filter and line impedance parameters on the stability of a MG in grid-connected mode.
- Sensitivity analysis of MG parameters was performed to investigate the most sensitive parameters that affect the SSS of a MG in grid-connected mode.

This paper is organized as follows: Section 2 presents the MG system and its parameters. Section 3 presents the state space of the MG system. Section 4 reports the simulation model that was modelled in MATLAB, while Section 5 shows the analysis and results of the MG model.

## 2. Proposed Controller Design

A MG supporting a three-phase grid with the proposed control method is represented in Figure 1, and the system parameters are detailed in Table 1. The figure shows that the MG is in grid-connected mode. The input is a DC voltage that is converted to AC through the inverter. However, the output voltage of the inverter is mostly a square wave, which requires a filter to convert the output square wave into a sine wave. The LCL filter is mainly used to interface the MG with the grid and to reduce the ripples produced by the high switching frequency of the inverter.

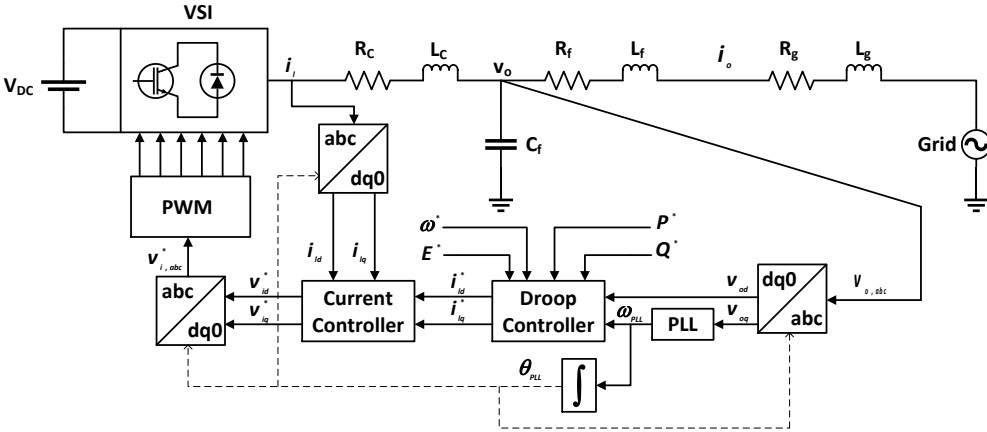

**Figure 1.** Three-phase voltage source inverter connected to a grid via a control method and a LCL filter.

**Table 1.** Parameters of the grid-tied inverter system [26].

| Parameter | Symbol | Value |
|---|---|---|
| Inverter-side parasitic resistance | $R_c$ | 0.25 Ω |
| Grid-side parasitic resistance of the LCL filter | $R_f$ | 0.25 Ω |
| Transmission line parasitic resistance | $R_g$ | 0.25 Ω |
| Inverter-side inductance of the LCL filter | $L_c$ | 4.77 mH |
| Grid-side inductance of the LCL filter | $L_f$ | 0.16 mH |
| Transmission line inductance | $L_g$ | 3.44 μH |
| Filter capacitance | $C_f$ | 14.7 μF |
| Proportional gain of the current controller | $k_{pc}$ | 14.24 |
| Integral gain of the current controller | $k_{ic}$ | $2.11 \times 10^4$ |
| Frequency droop gain | m | $1 \times 10^{-6}$ |
| Voltage droop gain | n | $1 \times 10^{-6}$ |
| Inverter input DC voltage | $v_{dc}$ | 180 V |
| Switching frequency | $f_s$ | 8 kHz |
| Grid voltage magnitude | $v_g$ | 120 V |
| Fundamental frequency | $f_n$ | 50 Hz |

A three-phase voltage and current should be transformed into a two-phase orthogonal rotating reference frame before entering a control system. The control system is working in the rotating reference frame (dq) and is then transferred back to the three-phase (abc) before entering the VSI, to appear at the input of the LCL filter. The reference active and reactive power are considered as inputs to the control system, where the inverter output power is controlled by controlling the inverter output current, while the inverter output voltage should be fixed. The active power reference is set to 500 W, while the reactive power is

set to zero, in order to make the power factor of the MG be at unity. Therefore, the total reactive power demand should be covered by the grid.

The droop controller is used to generate the reference currents based on the input reference values of the power. Therefore, the droop controller is responsible for meeting any loads connected to the MG. The current controller is a PI controller, which is responsible for comparing between the measured inverter output currents and the reference currents produced by the droop controller. The reference currents are adjusted by the PCC voltage and power references. The output of the PI current controller is a voltage that is fed to the PWM generator to be compared with the carrier signal, to generate the appropriate switching sequence and to allow controlling the voltage at the output of the inverter.

A phase locked loop (PLL) is used to decrease the frequency deviation between the grid and MG. Frequency deviation can severely affect the power sharing accuracy of a MG in grid-connected mode. The PLL is responsible for tracking the phase angle of the grid with the phase angle of the voltage at the PCC. Integrating the output frequency of the PLL provides the reference phase angle that is required for the transformation blocks.

## 3. State Space Model

To study the small signal stability of the system shown in Figure 1, the system is divided into five submodules. Each submodule is linearized and represented in the state space form. The general form of state space models is provided in [5,8]. The A matrix, which is the state matrix, of the whole system is formed by the combination of all of the submodules on a common reference frame. The stability was tested by analyzing the eigenvalues resulting from the combined system A matrix.

### 3.1. Phase-Locked Loop

The MG has to be synchronized with the main grid, as a lack of synchronization may cause large transient currents to occur at the point of common coupling (PCC). To ensure the synchronization process, the grid voltage amplitude, phase angle, and frequency should be accurately matched. This process is performed by the PLL.

The PLL in Figure 2 is used to track the MG frequency with the grid frequency and provide the transformation angle. The dynamics of the PLL are represented in (1)–(3). The small signal linearized state space model of the PLL is represented in (4) and (5).

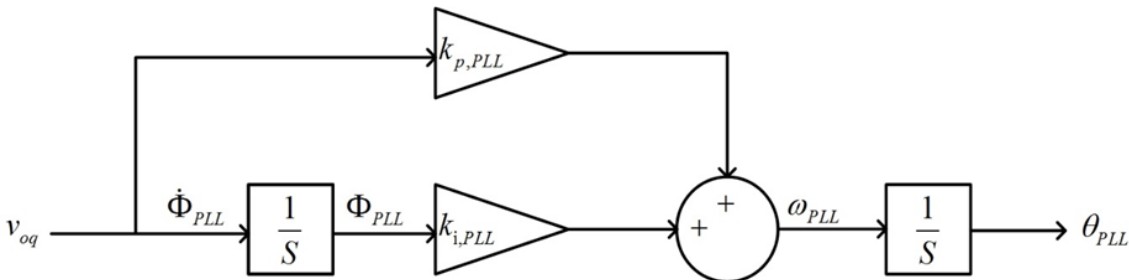

**Figure 2.** PLL.

It is assumed that $\Phi_{PLL} = \int v_{oq} dt$ to simplify the generation of the state-space model.

$$\dot{\varnothing}_{PLL} = k_{p,PLL} v_{oq} + k_{i,PLL} \Phi_{PLL} \tag{1}$$

$$\dot{\Phi}_{PLL} = v_{oq} \tag{2}$$

$$\omega_{PLL} = k_{p,PLL} v_{oq} + k_{i,PLL} \Phi_{PLL} \tag{3}$$

$$\begin{bmatrix} \Delta\dot{\varnothing}_{PLL} \\ \Delta\dot{\Phi}_{PLL} \end{bmatrix} = A_{PLL} \begin{bmatrix} \Delta\varnothing_{PLL} \\ \Delta\Phi_{PLL} \end{bmatrix} + B_{PLL} \begin{bmatrix} \Delta i_{ldq} \\ \Delta v_{odq} \\ \Delta i_{odq} \end{bmatrix} \tag{4}$$

$$[\Delta\omega_{PLL}] = [C_{PLL}]\begin{bmatrix} \Delta\varnothing_{PLL} \\ \Delta\Phi_{PLL} \end{bmatrix} + D_{PLL}\begin{bmatrix} \Delta i_{ldq} \\ \Delta v_{odq} \\ \Delta i_{odq} \end{bmatrix} \tag{5}$$

where

$$A_{PLL} = \begin{bmatrix} 0 & k_{i,PLL} \\ 0 & 0 \end{bmatrix}, B_{PLL} = \begin{bmatrix} 0 & 0 & 0 & k_{p,PLL} & 0 & 0 \\ 0 & 0 & 0 & 1 & 0 & 0 \end{bmatrix}$$

$$C_{PLL} = \begin{bmatrix} 0 & k_{i,PLL} \end{bmatrix}, D_{PLL} = \begin{bmatrix} 0 & 0 & 0 & k_{p,PLL} & 0 & 0 \end{bmatrix}$$

### 3.2. Droop Controller

Figure 3 shows the power control method, where the reference frequency is compared with the output frequency of the PLL and the error signal is passed through the droop frequency gain and added to the commanded active power. On the other side, the d-component of the inverter output voltage at the PCC is compared with the reference voltage, and the error signal is passed through the droop voltage gain and added to the commanded reactive power. The outputs of the droop controller are the current reference values. The dynamics of the droop controller are represented in (6)–(9). The small signal linearized state space model of the droop controller is represented in (10) and (11).

$$\dot{P} = P^* + m(\omega^* - \omega_{PLL}) \tag{6}$$

$$\dot{Q} = Q^* + n(E^* - v_{od}) \tag{7}$$

$$i_{ld}^* = \frac{2}{3v_{od}}\dot{P} = \frac{2}{3v_{od}}(P^* + m(\omega^* - \omega_{PLL})) \tag{8}$$

$$i_{lq}^* = -\frac{2}{3v_{od}}\dot{Q} = -\frac{2}{3v_{od}}(Q^* + n(E^* - v_{od})) \tag{9}$$

$$\begin{bmatrix} \Delta\dot{P} \\ \Delta\dot{Q} \end{bmatrix} = A_D\begin{bmatrix} \Delta P \\ \Delta Q \end{bmatrix} + B_{D1}\begin{bmatrix} \Delta i_{ldq} \\ \Delta v_{odq} \\ \Delta i_{odq} \end{bmatrix} + B_{D2}[\Delta\omega_{PLL}] \tag{10}$$

$$\begin{bmatrix} \Delta i_{ld}^* \\ \Delta i_{lq}^* \end{bmatrix} = C_D\begin{bmatrix} \Delta P \\ \Delta Q \end{bmatrix} + D_{D1}\begin{bmatrix} \Delta i_{ldq} \\ \Delta v_{odq} \\ \Delta i_{odq} \end{bmatrix} + D_{D2}[\Delta\omega_{PLL}] \tag{11}$$

where

$$A_D = \begin{bmatrix} 0 & 0 \\ 0 & 0 \end{bmatrix}, B_{D1} = \begin{bmatrix} 0 & 0 & 0 & 0 & 0 & 0 \\ 0 & 0 & -n & 0 & 0 & 0 \end{bmatrix}$$

$$B_{D2} = \begin{bmatrix} -m \\ 0 \end{bmatrix}, C_D = \begin{bmatrix} 0 & 0 \\ 0 & 0 \end{bmatrix}$$

$$D_{D1} = \begin{bmatrix} 0 & 0 & -\frac{2}{3v_{od}^2}(P^* + m(\omega^* - \omega_{PLL})) & 0 & 0 & 0 \\ 0 & 0 & \frac{2}{3v_{od}^2}(Q^* + n(E^*)) & 0 & 0 & 0 \end{bmatrix}$$

$$D_{D2} = \begin{bmatrix} -\frac{2m}{3v_{od}} \\ 0 \end{bmatrix}$$

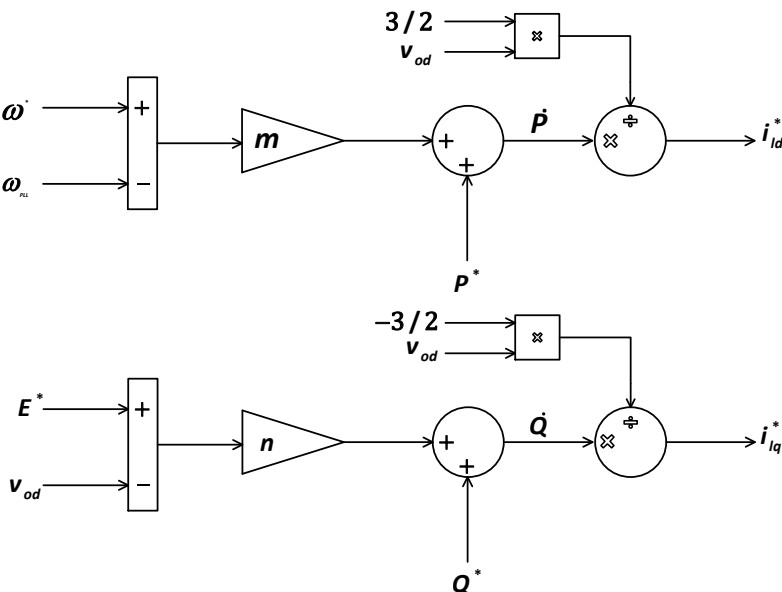

**Figure 3.** Droop controller.

### 3.3. Current Controller

Two PI current controllers are shown in Figure 4, and they are used to provide the output voltage of the inverter. The controllers compare the measured currents at the coupling inductor and the reference currents, which are the outputs of the previous submodule. The dynamics of the current controller are represented in (12)–(15). The small signal linearized state space model of the current controller is represented in (16) and (17).

$$\dot{\gamma}_d = i_{ld}^* - i_{ld} \tag{12}$$

$$\dot{\gamma}_q = i_{lq}^* - i_{lq} \tag{13}$$

$$v_{id}^* = -\omega_n L_c i_{lq} + k_{pc}(i_{ld}^* - i_{ld}) + k_{ic}\gamma_d \tag{14}$$

$$v_{iq}^* = \omega_n L_c i_{ld} + k_{pc}(i_{lq}^* - i_{lq}) + k_{ic}\gamma_q \tag{15}$$

$$\begin{bmatrix} \Delta\dot{\gamma}_d \\ \Delta\dot{\gamma}_q \end{bmatrix} = A_C \begin{bmatrix} \Delta\gamma_d \\ \Delta\gamma_q \end{bmatrix} + B_{C1} \begin{bmatrix} \Delta i_{ld}^* \\ \Delta i_{lq}^* \end{bmatrix} + B_{C2} \begin{bmatrix} \Delta i_{ldq} \\ \Delta v_{odq} \\ \Delta i_{odq} \end{bmatrix} \tag{16}$$

$$\begin{bmatrix} \Delta v_{id}^* \\ \Delta v_{iq}^* \end{bmatrix} = C_C \begin{bmatrix} \Delta\gamma_d \\ \Delta\gamma_q \end{bmatrix} + D_{C1} \begin{bmatrix} \Delta i_{ld}^* \\ \Delta i_{lq}^* \end{bmatrix} + D_{C2} \begin{bmatrix} \Delta i_{ldq} \\ \Delta v_{odq} \\ \Delta i_{odq} \end{bmatrix} \tag{17}$$

where

$$A_C = \begin{bmatrix} 0 & 0 \\ 0 & 0 \end{bmatrix}, B_{C1} = \begin{bmatrix} 1 & 0 \\ 0 & 1 \end{bmatrix}$$

$$B_{C2} = \begin{bmatrix} -1 & 0 & 0 & 0 & 0 & 0 \\ 0 & -1 & 0 & 0 & 0 & 0 \end{bmatrix}, C_C = \begin{bmatrix} k_{ic} & 0 \\ 0 & k_{ic} \end{bmatrix}$$

$$D_{C1} = \begin{bmatrix} k_{pc} & 0 \\ 0 & k_{pc} \end{bmatrix}, D_{C2} = \begin{bmatrix} -k_{pc} & -\omega_n L_c & 0 & 0 & 0 & 0 \\ \omega_n L_c & -k_{pc} & 0 & 0 & 0 & 0 \end{bmatrix}$$

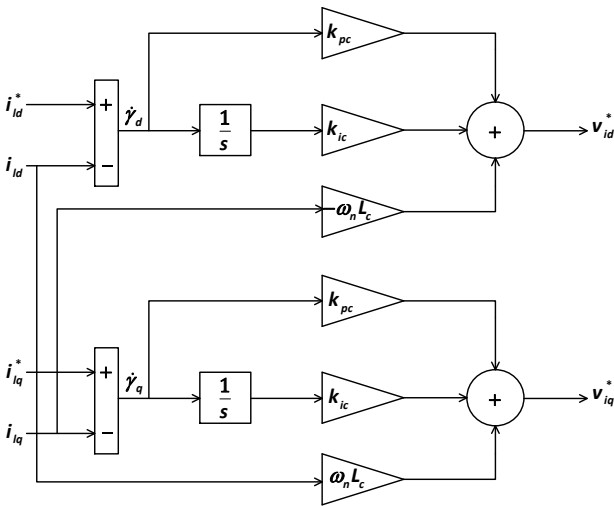

**Figure 4.** Current controller.

### 3.4. LCL Filter

The switches of the inverter are assumed to be ideal and they produce no losses, which makes the inverter voltage the same as the demanded voltage $v_i = v_i^*$. The dynamics of the LCL filter with the feeder are represented in (18)–(23). The PLL frequency, which is the inverter frequency, is considered to be the common reference frame. Therefore, all of the equations are represented in the inverter reference frame, except that the voltage at the grid bus needs to converted according to the transformation matrix, as in (24) and (25). The grid voltage is transformed to the common reference frame according to (26), where the transformation angle δ represents the phase angle difference between the grid voltage and the inverter voltage. Finally, the equations are linearized and represented in the state-space form. The small signal linearized state space model of the LCL filter is represented in (27).

$$\dot{i}_{ld} = -\frac{R_c}{L_c}i_{ld} + \omega_{PLL}i_{lq} + \frac{1}{L_c}v_{id} - \frac{1}{L_c}v_{od} \tag{18}$$

$$\dot{i}_{lq} = -\frac{R_c}{L_c}i_{lq} - \omega_{PLL}i_{ld} + \frac{1}{L_c}v_{iq} - \frac{1}{L_c}v_{oq} \tag{19}$$

$$\dot{v}_{od} = \omega_{PLL}v_{oq} + \frac{1}{C_f}i_{ld} - \frac{1}{C_f}i_{od} \tag{20}$$

$$\dot{v}_{oq} = \omega_{PLL}v_{od} + \frac{1}{C_f}i_{lq} - \frac{1}{C_f}i_{oq} \tag{21}$$

$$\dot{i}_{od} = -\frac{(R_g + R_f)}{L_g + L_f}i_{od} + \omega_{PLL}i_{oq} + \frac{1}{L_g + L_f}(v_{od} - v_{gd}) \tag{22}$$

$$\dot{i}_{oq} = -\frac{(R_g + R_f)}{L_g + L_f}i_{oq} - \omega_{PLL}i_{od} + \frac{1}{L_g + L_f}(v_{oq} - v_{gq}) \tag{23}$$

$$f^{DQ,G} = T_{global}^{local} f^{dq,g} \tag{24}$$

$$T_{global}^{local} = \begin{bmatrix} \cos(\delta) & -\sin(\delta) \\ \sin(\delta) & \cos(\delta) \end{bmatrix} \ , \ \delta = \int (\omega - \omega_{PLL}) \tag{25}$$

$$\begin{aligned} v_{gd} &= V_{gD}\cos(\delta_1) - V_{gQ}\sin(\delta_1) \\ v_{gq} &= V_{gD}\sin(\delta_1) + V_{gQ}\cos(\delta_1) \end{aligned} \tag{26}$$

where $f^{DQ, G}$ is the global frequency, $f^{dq,g}$ is the local frequency, and $V_{g,DQ}$ is the grid bus voltage in the global reference frame.

$$
\begin{bmatrix} \Delta \dot{i}_{ldq} \\ \Delta \dot{v}_{odq} \\ \Delta \dot{i}_{odq} \end{bmatrix} = A_{LCL} \begin{bmatrix} \Delta i_{ldq} \\ \Delta v_{odq} \\ \Delta i_{odq} \end{bmatrix} + B_{LCL1} \begin{bmatrix} \Delta v_{id} \\ \Delta v_{iq} \end{bmatrix} + B_{LCL2}[\Delta \delta_1] + B_{LCL3}[\Delta \omega_{PLL}] \quad (27)
$$

where

$$
A_{LCL} = \begin{bmatrix} -\frac{R_c}{L_c} & \omega_{PLL} & -\frac{1}{L_c} & 0 & 0 & 0 \\ -\omega_{PLL} & -\frac{R_c}{L_c} & 0 & -\frac{1}{L_c} & 0 & 0 \\ \frac{1}{C_f} & 0 & 0 & \omega_{PLL} & -\frac{1}{C_f} & 0 \\ 0 & \frac{1}{C_f} & -\omega_{PLL} & 0 & 0 & -\frac{1}{C_f} \\ 0 & 0 & \frac{1}{L_g+L_f} & 0 & -\frac{(R_g+R_f)}{L_g+L_f} & \omega_{PLL} \\ 0 & 0 & 0 & \frac{1}{L_g+L_f} & -\omega_{PLL} & -\frac{(R_g+R_f)}{L_g+L_f} \end{bmatrix}
$$

$$
B_{LCL1} = \begin{bmatrix} \frac{1}{L_c} & 0 \\ 0 & \frac{1}{L_c} \\ 0 & 0 \\ 0 & 0 \\ 0 & 0 \\ 0 & 0 \end{bmatrix}, \ B_{LCL2} = \begin{bmatrix} 0 \\ 0 \\ 0 \\ 0 \\ \frac{V_{gD} \sin(\delta_1)}{L_g+L_f} \\ -\frac{V_{gD} \cos(\delta_1)}{L_g+L_f} \end{bmatrix}, \ B_{LCL3} = \begin{bmatrix} i_{lq} \\ -i_{ld} \\ v_{oq} \\ -v_{od} \\ i_{oq} \\ -i_{od} \end{bmatrix}
$$

*3.5. Bus*

The grid angular frequency is represented by $\omega_g$, and the inverter angular frequency is $\omega_{PLL}$. The phase difference between the grid and the inverter is $\delta_1$, which is used to transform the grid voltage from the global frame to the local frame. This angle varies consistently with the variation of the inverter frequency, as described in (28) and (29). The small signal linearized state space model of the bus is represented in (30).

$$
\dot{\delta}_1 = \omega_g - \omega_{PLL} \quad (28)
$$

$$
\dot{\delta}_1 = \omega_g - k_{p,PLL} v_{oq} - k_{i,PLL} \Phi_{PLL} \quad (29)
$$

$$
\left[ \Delta \dot{\delta}_1 \right] = A_{BUS}[\Delta \delta_1] + B_{BUS}[\Delta \omega_{com}] \quad (30)
$$

where

$$
A_{BUS} = [0], \ B_{BUS} = [-1]
$$

*3.6. Full System*

The combined submodules are represented in (31) where there are 13 states and five inputs. The $A_{sys}$ represents the state matrix of the full model, and $B_{sys}$ represents the input matrix to the system.

$$
[\Delta \dot{x}_{sys}] = A_{sys}[\Delta x_{sys}] + B_{sys}[\Delta u_{sys}] \quad (31)
$$

$$
u = \begin{bmatrix} \omega_{PLL} & P^* & Q^* & V_{gD} & V_{gQ} \end{bmatrix}^T
$$

$$
x = \begin{bmatrix} \varnothing_{PLL} & \Phi_{PLL} & P & Q & \gamma_d & \gamma_q & i_{ld} & i_{lq} & v_{od} & v_{oq} & i_{od} & i_{oq} & \delta_1 \end{bmatrix}^T
$$

$$A_{sys} = \begin{bmatrix} A_{PLL} & 0 & 0 & B_{PLL} & 0 \\ B_{D2}C_{PLL} & A_D & 0 & B_{D1} + B_{D2}D_{PLL} & 0 \\ B_{C1}D_{D2}C_{PLL} & B_{C1}C_D & A_C & B_{C2} + B_{C1}D_{D1} + B_{C1}D_{D2}D_{PLL} & 0 \\ B_{LCL3}C_{PLL} + B_{LCL1}D_{C1}D_{D2}C_{PLL} & B_{LCL1}D_{C1}C_D & B_{LCL1}C_C & A_{LCL} + B_{LCL1}D_{C2} + B_{LCL1}D_{C1}D_{D1} + B_{LCL1}D_{C1}D_{D2}D_{PLL} + B_{LCL3}D_{PLL} & B_{LCL2} \\ B_{BUS}C_{PLL} & 0 & 0 & B_{BUS}D_{PLL} & A_{BUS} \end{bmatrix}_{13\times13}$$

$$A_{sys} = \begin{bmatrix}
0 & K_{i,PLL} & 0 & 0 & 0 & 0 & 0 & 0 & 0 & K_{p,PLL} & 0 & 0 & 0 \\
0 & 0 & 0 & 0 & 0 & 0 & 0 & 0 & 0 & 1 & 0 & 0 & 0 \\
0 & -K_{i,PLL}\cdot m & 0 & 0 & 0 & 0 & 0 & 0 & 0 & -K_{p,PLL}\cdot m & 0 & 0 & 0 \\
0 & 0 & 0 & 0 & 0 & 0 & 0 & -n & 0 & 0 & 0 & 0 & 0 \\
0 & -\dfrac{2\cdot K_{i,PLL}\cdot m}{3\cdot v_{od}} & 0 & 0 & 0 & -1 & 0 & \dfrac{-(2\cdot(P^*-m(\omega_{PLL}-\omega^*)))}{3\cdot v_{od}^2} & -\dfrac{2\cdot K_{p,PLL}\cdot m}{3\cdot v_{od}} & 0 & 0 & 0 \\
0 & 0 & 0 & 0 & 0 & 0 & -1 & \dfrac{2\cdot(Q^*+(E^*\cdot n))}{3\cdot v_{od}^2} & 0 & 0 & 0 & 0 & 0 \\
0 & K_{i,PLL}\cdot i_{lq}-\dfrac{2\cdot K_{i,PLL}\cdot K_{pc}\cdot m}{3\cdot L_c\cdot v_{od}} & 0 & 0 & \dfrac{K_{ic}}{L_c} & 0 & \dfrac{-K_{pc}}{L_c}-\dfrac{R_c}{L_c} & \omega_{PLL}-\omega_n & -\dfrac{1}{L_c}-\dfrac{(2\cdot K_{pc}\cdot(P^*-m(\omega_{PLL}-\omega^*)))}{3\cdot L_c\cdot v_{od}^2} & K_{p,PLL}\cdot i_{lq}-\dfrac{2\cdot K_{pc}\cdot K_{p,PLL}\cdot m}{3\cdot L_c\cdot v_{od}} & 0 & 0 & 0 \\
0 & -K_{i,PLL}\cdot i_{ld} & 0 & 0 & 0 & \dfrac{K_{ic}}{L_c} & \omega_n-\omega_{PLL} & \dfrac{-K_{pc}}{L_c}-\dfrac{R_c}{L_c} & \dfrac{2\cdot K_{pc}\cdot(Q^*+(E^*\cdot n))}{3\cdot L_c\cdot v_{od}^2} & -K_{p,PLL}\cdot i_{ld}-\dfrac{1}{L_c} & 0 & 0 & 0 \\
0 & K_{i,PLL}\cdot v_{oq} & 0 & 0 & 0 & 0 & \dfrac{1}{C_f} & 0 & 0 & \omega_{PLL}+(K_{p,PLL}\cdot v_{oq}) & -\dfrac{1}{C_f} & 0 & 0 \\
0 & -K_{i,PLL}\cdot v_{od} & 0 & 0 & 0 & 0 & \dfrac{1}{C_f} & -\omega_{PLL} & 0 & -K_{p,PLL}\cdot v_{od} & 0 & -\dfrac{1}{C_f} & 0 \\
0 & K_{i,PLL}\cdot i_{oq} & 0 & 0 & 0 & 0 & 0 & \dfrac{1}{L_g+L_f} & 0 & K_{p,PLL}\cdot i_{oq} & -\dfrac{R_g+R_f}{L_g+L_f} & \omega_{PLL} & \dfrac{V_{gD}\sin(\delta_1)}{L_g+L_f} \\
0 & -K_{i,PLL}\cdot i_{od} & 0 & 0 & 0 & 0 & 0 & 0 & \dfrac{1}{L_g+L_f} & \dfrac{1}{L_g+L_f}-(K_{p,PLL}\cdot i_{od}) & -\omega_{PLL} & -\dfrac{R_g+R_f}{L_g+L_f} & -\dfrac{V_{gD}\cos(\delta_1)}{L_g+L_f} \\
0 & -K_{i,PLL} & 0 & 0 & 0 & 0 & 0 & 0 & 0 & -K_{p,PLL} & 0 & 0 & 0
\end{bmatrix}_{13\times13}$$

## 4. Simulation Model

The MG system was modelled in MATLAB Simulink, as shown in Figure 5, and the model parameters are reported in Table 1. The solver type in MATLAB was set to ode45x with a time step of 1 µs. The total simulation time was eighteen minutes and it was performed using a personal computer with an Intel® Core™ i7-8750H CPU @ 2.2 GHz processor and memory (RAM) of 16.00 GB.

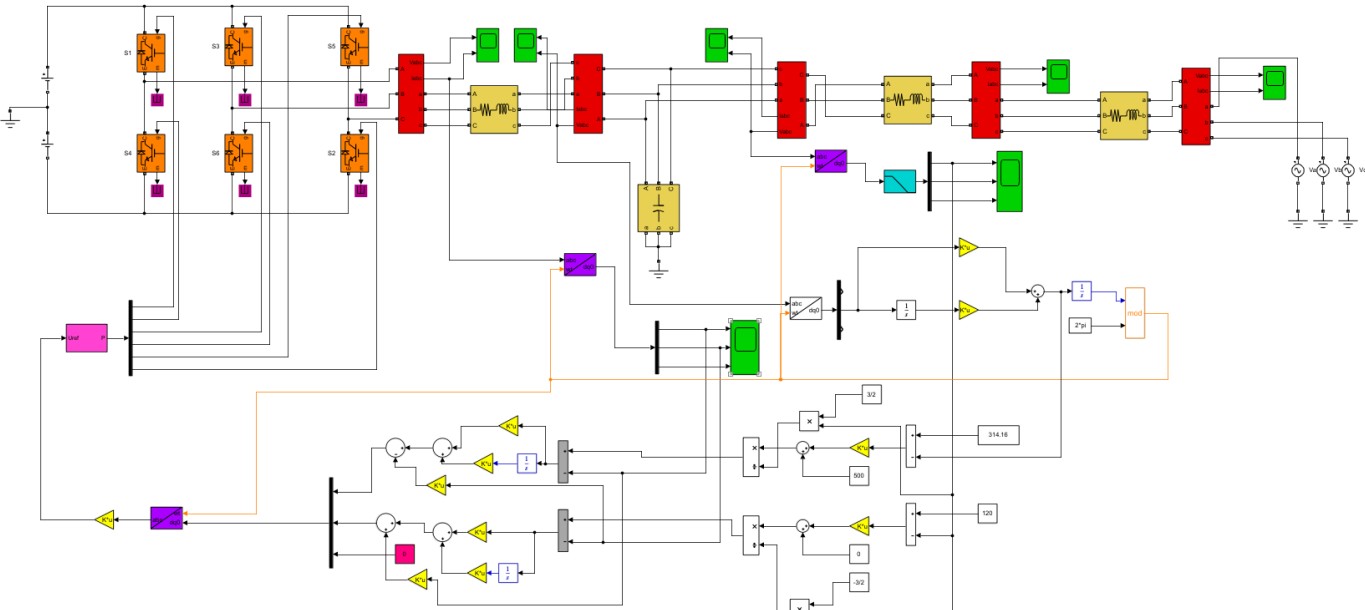

**Figure 5.** Screen capture of a MG system in grid-connected mode modelled in MATLAB Simulink.

## 5. Results and Discussion

Figure 6 shows the active and reactive power output of the MG where they match the reference values of 500 W and 0 VAR, respectively. Small perturbations in the parameter values of any system may change the system's behavior significantly, causing instability. The effect of parameter variation on a specific system behavior can be determined by calculating the parameter sensitivity. The parameter sensitivity analysis gives information about parameters that can cause instability with only small changes in their values. Therefore,

parameter values can be designed carefully to avoid instability. The parameter sensitivity is calculated by

$$Sen_{pj}^{\lambda_i} = \frac{\left|\lambda_i(p + \Delta p_j) - \lambda_i(p)\right|}{\Delta p_j} \tag{32}$$

where $\lambda_i(p)$ is the ith eigenvalue of the system before perturbation, $\lambda_i(p + \Delta p_j)$ is the ith eigenvalue of the system after perturbation, and $\Delta p_j$ is the difference between the original parameter value and the perturbed parameter value. The parameter sensitivity analysis is illustrated in Figure 7, which shows that the most sensitive parameters in the MG system were $L_g$ and $L_f$. Therefore, it was decided to study the impact of changing their values on the stability condition.

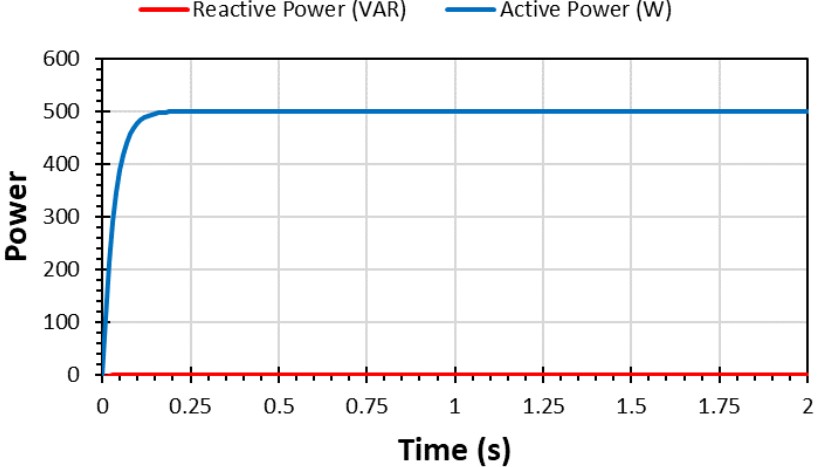

**Figure 6.** Output power of the MG.

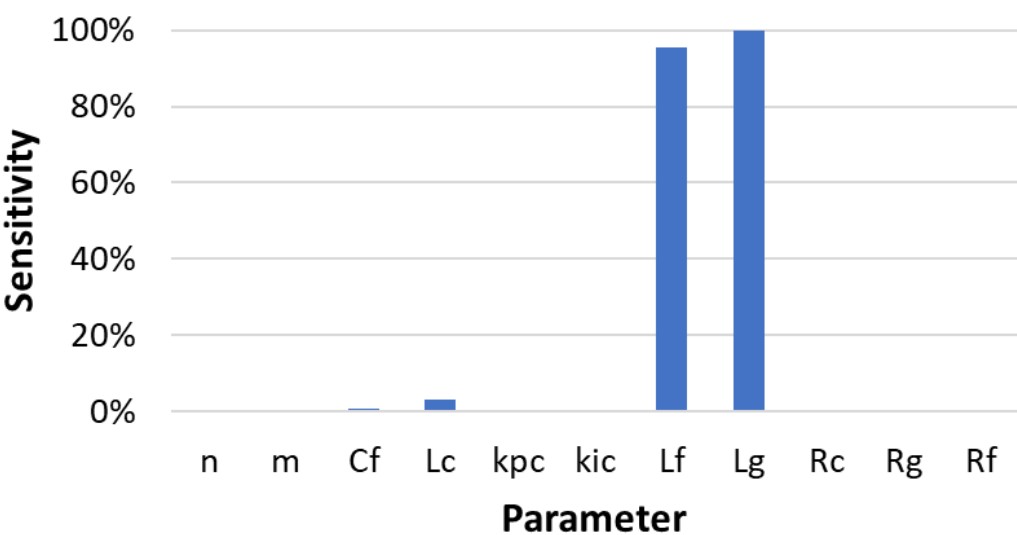

**Figure 7.** Parameter sensitivity analysis of the MG system.

The eigenvalues of the combined small signal model can be related to the system parameters and states. This relationship can help us to optimize the system dynamic response and stability. The trace of the eigenvalues when changing the grid side inductance of the LCL filter is shown in Figure 8. The inductance value was varied in steps of 0.3 mH in the time domain simulation performed by MATLAB. The figure shows that the value of $L_f$ that destabilized the system was 6.2 mH. In order to assess the accuracy of the mathematical model, it was decided to simulate the system with different values of $L_f$ and measure the d-component of the voltage at the PCC. It was found in the simulation that the system became unstable at 6.2 mH of $L_f$, as shown in Figure 9. The figure also shows that when

reducing the $L_f$ value by 0.3 mH, the system was stable, while the system was unstable when it was increased by 0.3 mH.

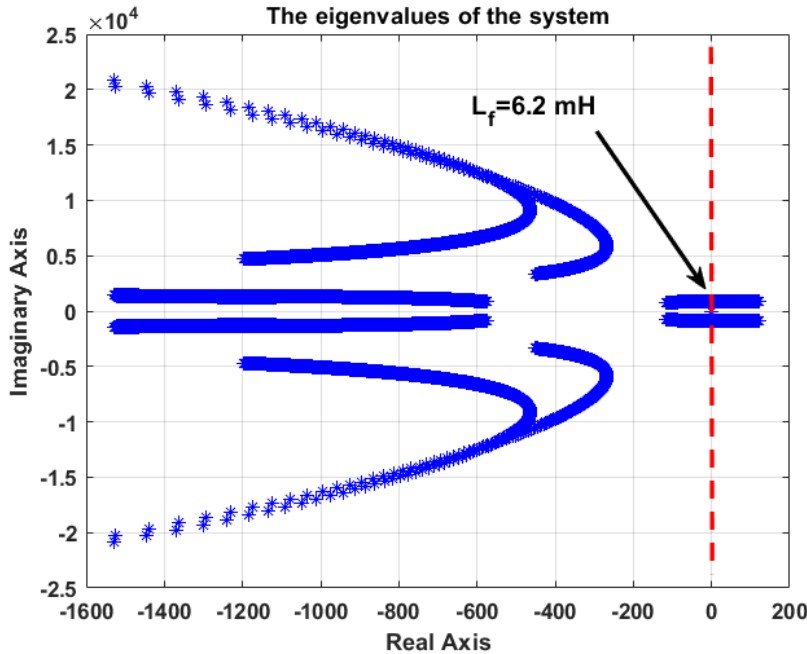

**Figure 8.** Trace of the eigenvalues of the linearized mathematical model under the variation of $L_f$ in the range of 0.16 mH and 0.01 H.

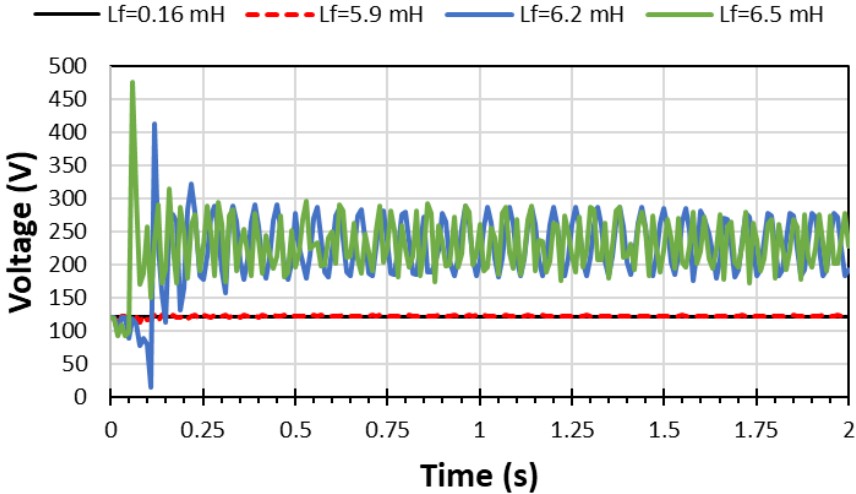

**Figure 9.** Simulation results of the voltage at the PCC when varying $L_f$.

When varying the transmission line inductance in the mathematical model in steps of 0.3 mH, it was found that the system became unstable at 6.1 mH, as shown in Figure 10. This value is validated by the simulation results shown in Figure 11. The figure shows that when applying 6.1 mH and above, the system is unstable, with the appearance of high voltage values with oscillations. When reducing the inductance value by 0.3 mH and when applying the original setting of 3.44 μH, the system was stable, and the voltage stayed around 120 V, without any high oscillations.

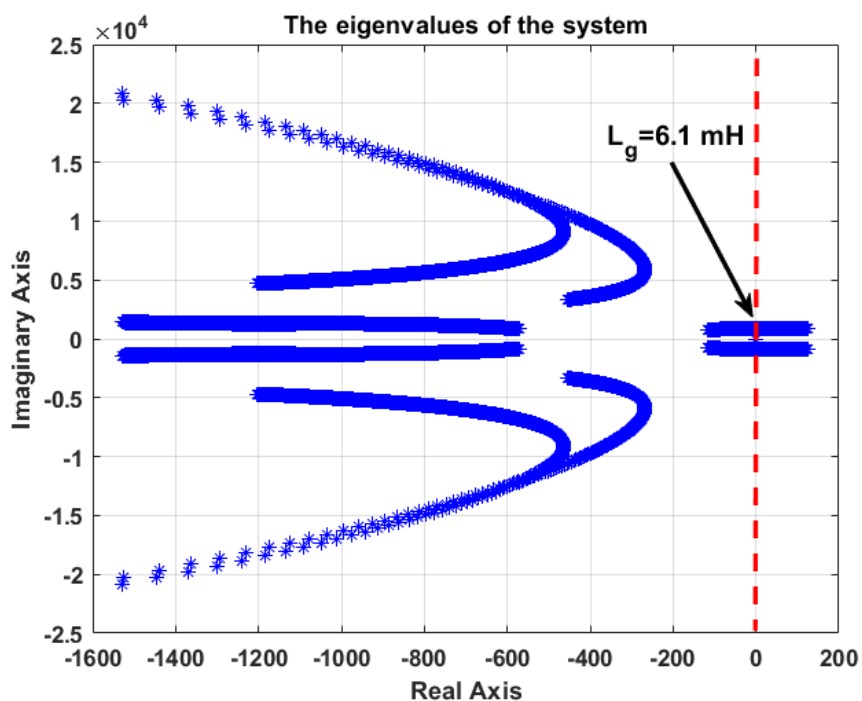

**Figure 10.** Trace of the eigenvalues of the linearized mathematical model under the variation of $L_g$ in the range of 3.44 µH and 0.01 H.

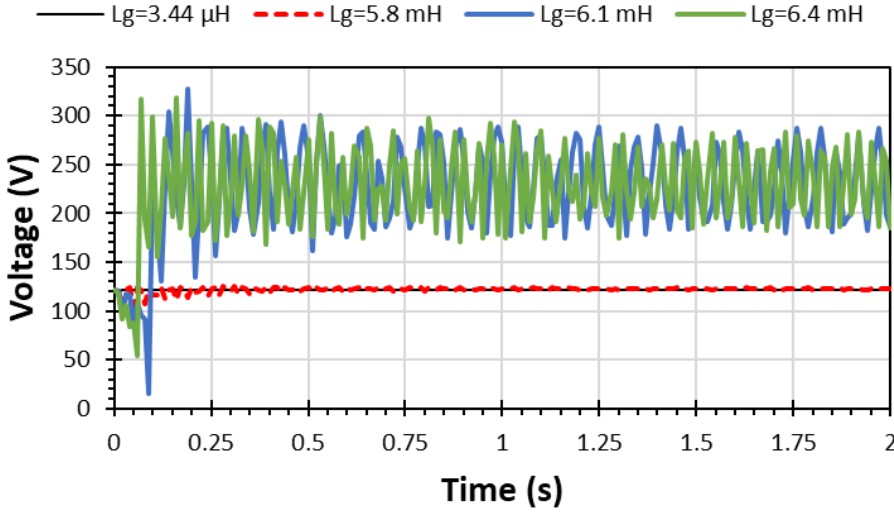

**Figure 11.** Simulation results of the voltage at the PCC when varying $L_g$.

The inverter-side inductance of the LCL filter did not affect the stability of the system, as shown in Figure 12. This mathematical result is validated by the simulation results shown in Figure 13. The figure shows that the system was stable, despite the significant increase in the inductance value. These results are confirmed by the parameter sensitivity analysis shown in Figure 7, which shows that the MG system was not highly sensitive to changes in the inverter-side inductance.

The parameter sensitivity analysis showed that the poorly damped modes were related to the LCL filter parameters and the feeder inductance. Consequently, the damping ratio of these modes can be increased by increasing the series damping resistances of these sensitive parameters. However, passive damping has some drawbacks, such as increasing the power loss in the filter and weakening the ability of the LCL filter to attenuate the harmonics generated by the switching in the inverter. Furthermore, it is not possible to increase the damping of the feeder because, in practice, this represents the distance between

the MG and grid. Therefore, active damping should be designed to avoid decreasing the system's efficiency.

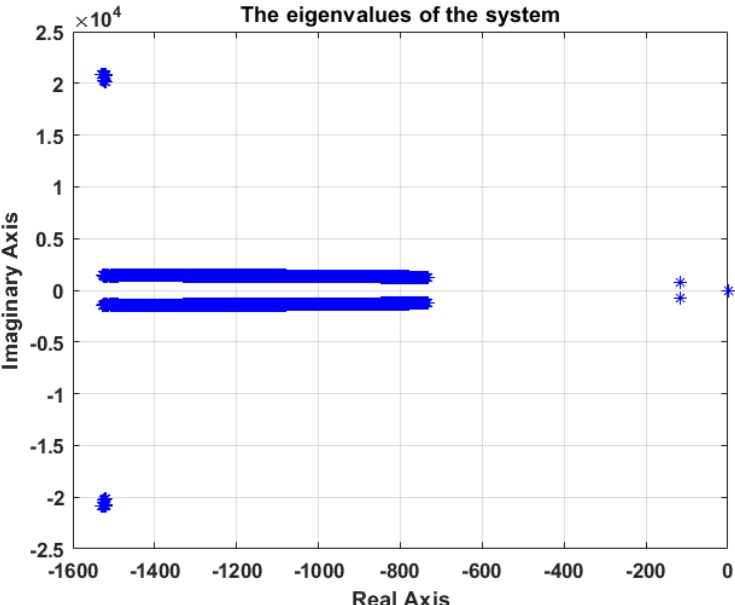

**Figure 12.** Trace of the eigenvalues of the linearized mathematical model under the variation of $L_c$ in the range of 4.77 mH and 0.01 H.

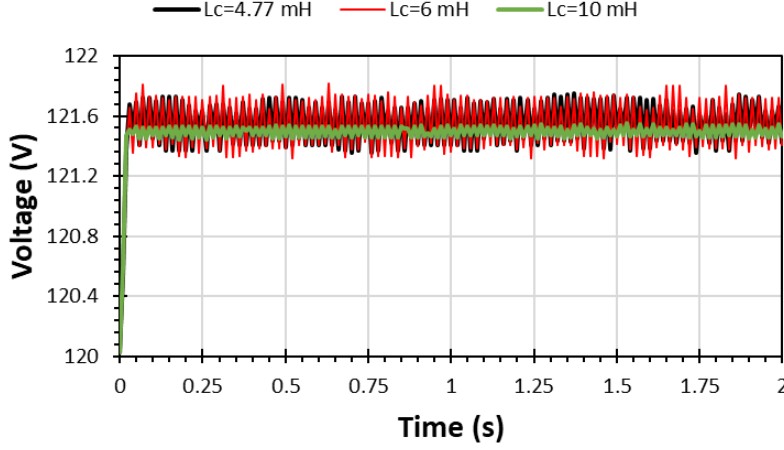

**Figure 13.** Simulation results of the voltage at the PCC when varying $L_c$.

## 6. Conclusions

SSS has been discussed in this article for a MG in grid-connected mode, to improve the control design and also the stability of the system. A controller proposal, to control a MG in grid-connected mode, was introduced in this paper. The whole system of the MG, including the control system, was divided into submodules, and the small signal model for each submodule was produced in detail. The small signal models of the submodules were combined together on the inverter reference frame, to generate the A matrix of the full system, which was used to analyze the stability of the full system. A sensitivity analysis of the MG parameters was performed, before applying small variations to the most sensitive parameters, to the test stability. The aim of this dynamic modelling was to design the MG parameters that ensure stability. It was found that increasing the inductance destabilizes the MG system. When increasing the grid side inductance of the LCL filter beyond the value of 6.2 mH, the system became unstable; and when increasing the transmission line inductance beyond the value of 6.1 mH, the system also became unstable. The small signal model

mathematical results were validated by comparison with the simulation results produced using MATLAB Simulink. Identifying stability margins is a crucial aspect of increasing grid resilience and of avoiding instability, which causes power outages and high costs, resulting from damages to the grid infrastructure. Furthermore, power outages may affect the economy indirectly by reducing sales and productivity. In future work, a solar power plant will be added, alongside electrical loads that represent the real loads of consumers, before studying the SSS, based on variations in the power source and electrical loads.

**Funding:** This research received no external funding.

**Institutional Review Board Statement:** Not applicable.

**Informed Consent Statement:** Not applicable.

**Data Availability Statement:** Not applicable.

**Acknowledgments:** The author would like to acknowledge Jouf University for supporting this research project.

**Conflicts of Interest:** The author declares no conflict of interest.

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
