# Peer review of "Small Signal Stability Analysis of a Microgrid in Grid-Connected Mode"

_sustainability, doi:10.3390/su14159372_

Round 1
Reviewer 1 Report
Paper implements a small signal stability analysis for grid-connected microgrid. Some comments can be found bellow:
-In the Introduction part, the authors need to clearly express the main contribution of the manuscript at the end of Introduction part by some outlines. And then present the paper’s organization at the end of the Introduction part.
-It confused me when I read t abstract an Introduction, I don’t know if you want to find the stability margin or you want to investigate the impact of the LCL filter and line impedance parameters? Please clarify and revise the abstract and Introduction accordingly. The reference papers are old, need to be updated with the state-of-the-art papers, more related work should be cited here:
https://www.mdpi.com/1996-1073/13/2/451
https://ieeexplore.ieee.org/document/9511198
https://link.springer.com/article/10.1007/s11432-021-3290-3
-Please mention the source of all equations if they are derived from a specific reference.
- The results analysis is just a brief review of the figures presented in only 20 lines. The authors should be more specific and explain the real findings of the paper by further analyzing the results and by commenting on them.
- What is the stability margin of the considered MG control system? Can that margin be applied the same for the other microgrid system?
- Conclusion is poor written.
Reviewer 2 Report
In this paper the authors study the problem of finding the stability margin of a Microgrid control system, line impedance, and passive filter parameters. Consistent power system operation is very important to both power utilities and consumers.
This paper proposes a comprehensive an approach based on Microgrid stability issues are classified into three categories that are transient, voltage and small signal stability.
The topic is quite interesting, but I have comments. The introduction should place the proposed approach on the background of existing and known solution presented in literature. Also the importance of the research field should be stressed.
In my opinion, this article lacks specifics. The well-known approach is outlined here. In this article, it is necessary to simulate a real electrical network to a solar power plant connected to it. To do this, it is necessary in the article to bring an electrical network with a voltage of 380 V with real loads of consumers. Connect a solar power plant to this network, for example, with a capacity of 5 or 10 kW. Then carry out simulation of transient processes that occur during the operation of a solar power plant and an electrical network. Then it will become clear to the reader why this study is being carried out in the article.
The presented report is at a very high scientific level. I believe that the present study has a significant scientific and applied contribution, which is strongly emphasized in the basically reporting volume. A slight clarification can be made in the abstract part, where the quality of the research can be enhanced. In the conclusions, it is necessary to describe what is the economic effect of using the approach proposed in the article?
Round 2
Reviewer 1 Report
The paper has been revised as commented. I have no further comments.
Reviewer 2 Report
Thanks to the authors, all my comments were taken into account. I recommend this article for publication.